# Appropriateness of indirect markers of muscle damage following lower limbs eccentric-biased exercises: A systematic review with meta-analysis

**Emeric Chalchat**[1,2]\*, **Anne-Fleur Gaston**[3], **Keyne Charlot**[1,4], **Luis Peñailillo**[5], **Omar Valdés**[6], **Pierre-Emmanuel Tardo-Dino**[1,4], **Kazunori Nosaka**[7], **Vincent Martin**[2,8], **Sebastian Garcia-Vicencio**[1,4©], **Julien Siracusa**[1,4©]

**1** Institut de Recherche Biomédicale des Armées, Unité de Physiologie des Exercices et Activités en Conditions Extrêmes, Département Environnements Opérationnels, Bretigny-Sur-Orge, France, **2** Université Clermont Auvergne, AME2P, Clermont-Ferrand, France, **3** Laboratoire Interdisciplinaire Performance Santé Environnement de Montagne, Université de Perpignan, Perpignan, France, **4** LBEPS, Univ Evry, IRBA, Université Paris Saclay, Evry, France, **5** Exercise and Rehabilitation Sciences Laboratory, School of Physical Therapy, Faculty of Rehabilitation Sciences, Universidad Andres Bello, Santiago, Chile, **6** Exercise Science Laboratory, School of Kinesiology, Faculty of Medicine, Universidad Finis Terrae, Santiago, Chile, **7** Centre for Exercise and Sports Science Research, School of Medical and Health Sciences, Edith Cowan University, Joondalup, WA, Australia, **8** Institut Universitaire de France (IUF), Paris, France

© These authors contributed equally to this work.

\* emchalchat@gmail.com

## Abstract

### Purpose

The aim of this review was to (1) characterize the time-course of markers of exercise-induced muscle damage (EIMD) based on the level of maximal voluntary contraction torque loss at 24-48h post-exercise ($MVC_{loss24-48h}$), (2) identify factors (e.g., exercise and population characteristics) affecting the level of $MVC_{loss24-48h}$, and (3) evaluate the appropriateness of EIMD markers as indicators of $MVC_{loss24-48h}$.

### Methods

Magnitude of change of each EIMD markers was normalized using the standardized mean differences method to compare the results from different studies. Time-course of EIMD markers were characterized according to three levels of $MVC_{loss24-48h}$ based on a clustering analysis of the 141 studies included. Association between $MVC_{loss24-48h}$ levels and participant´s characteristics or exercise type/modalities were assessed. Meta-regressions were performed to investigate the associations between $MVC_{loss24-48h}$ and EIMD markers changes at <6h, 24h, 48h, 72h and >96h after exercise.

### Results

Time-course of EIMD markers recovery differs between levels of $MVC_{loss24-48h}$. Training status and exercise type/modality were associated with $MVC_{loss24-48h}$ level ($p<0.05$).

**Data Availability Statement:** All relevant data are within the manuscript and its Supporting Information files.

**Funding:** The author(s) received no specific funding for this work.

**Competing interests:** The authors have declared that no competing interests exist.

$MVC_{loss24-48h}$ was correlated to changes in myoglobin concentration (<6h), jump height (24h) and range of motion (48h) ($p<0.001$).

## Conclusion

As the exercise could differently affect markers as function of the EIMD severity (i.e., $MVC_{loss24-48h}$ levels), different markers should be used as function of the timing of measurement. Mb concentration should be used during the first hours after the exercise (<6h), whereas jump height (24h) and range of motion (48h) could be used as surrogate for maximal voluntary contraction later. Moreover, training status and exercise type/modality could influence the magnitude of $MVC_{loss24-48h}$.

## Introduction

Eccentric biased exercise – where muscles are activated while lengthening – has received considerable interest in the field of health and sports sciences over the last decades. The use of progressive eccentric exercise has been proposed for the treatment of a variety of chronic diseases such as tendinopathies [1], coronary artery disease [2], knee osteoarthritis [3] or for the preservation of muscle mass in chronic obstructive pulmonary disease patients [4]. The implementation of eccentric contractions as a part of resistance training program has also been shown to be beneficial for the improvement of athletic performance through gain of muscle mass, strength, and power [5–7] and useful as an injury prevention/rehabilitation strategy [8]. Despite the benefits of this type of exercise, lengthening contractions can alter the ultrastructure of the muscle tissue and induce exercise-induced muscle damage (EIMD) [9].

EIMD is characterized by a prolonged reduction of muscle force and power production which has shown to better represent magnitude of muscle damage when assessed at 24-48h after exercise [10, 11]. The reduction in muscle function is often accompanied by other indirect markers of muscle damage, which have traditionally been used to quantify magnitude of EIMD that can last from few days to several weeks following exercise. Among indirect markers of muscle damage, the delayed-onset muscle soreness (DOMS), decreased joint range of motion (ROM) as well as increased blood levels of circulating muscle proteins such as creatine kinase (CK) and myoglobin (Mb) are the most frequently reported [10]. These EIMD markers are often measured concomitantly to describe a complex succession of events encompassing several physiological processes such as loss of myofibrillar integrity, connective tissue damage, membrane damage, failure in excitation-contraction coupling, and extracellular matrix remodeling [9–11].

Warren et al. [10], within the limitations of a narrative review approach, published recommendations in which the measurement of muscle strength (i.e., maximal voluntary contraction torque) was raised as the best indirect assessment of the magnitude of EIMD compared to other indirect markers due to its reliability, low cost, accessibility and easy to use [12]. Maximal voluntary contraction (MVC) torque comes closest to evaluating the overall functional contractile capacity of the muscle then has become a great tool in rehabilitation healthcare [12]. Despite the high set of indirect markers reported to assess muscle damage, their usefulness to quantify the magnitude and time-course of EIMD is still under debate, as their responses do not always converge. Since then, numerous studies highlighted an important variability of EIMD markers both in magnitude and time-course, which has limited the interpretation of such changes [11, 13]. Furthermore, as it has already been pointed [11, 13], the

discrepancies between markers responses may be partly explained by an important inter-individual variability (age, sex, training status, etc.), the limited sample size, nature of exercise, and muscle groups involved in exercise. Interestingly, a comprehensive synthesis using a systematic and quantitative approach has not been yet performed, making difficult the decision of what is (are) the best marker(s) to monitor EIMD. The evaluation of the appropriateness of the most reported indirect markers of EIMD could be helpful in a sport context to prevent other-related musculoskeletal injuries (e.g., primary lower extremity injury) in the days following the exercise. Furthermore, EIMD is not desired when a clinical population is performing exercise to improve health and adhesion and enjoyment are highly desired to induce long-lasting changes in patients. In this context, it may be of great interest to quantitatively summarize exercise parameters and population characteristics that could influence the magnitude and time course indirect markers of EIMD.

In an attempt to shed light in the complexity, we performed a systematic review and a meta-analysis with the aim to characterize the magnitude and time-course of commonly used indirect markers of EIMD over different levels of muscle function loss at 24–48 h post-exercise (based on the effect size values) by using a classification analysis (clustering) as previously suggested by Damas et al. [13]. The second aim was to identify factors such as exercise parameters and population characteristics affecting the magnitude of muscle strength loss after exercise. Finally, we aimed to evaluate the appropriateness of EIMD markers as indicators of muscle functional deficit.

## Materials and methods

### Literature search

Five databases including Pubmed, Embase, Web of Science, Cochrane and Scopus were searched using the following terms: ("muscle damage" OR "muscle injury" OR "EIMD") AND (contraction OR exercise) AND (strength OR MVC OR torque OR force OR "muscle function" OR "neuromuscular function" OR "maximal voluntary contraction") NOT ("mouse" OR "Mice" OR "rat" OR "animal") for "English-language" papers from start of records until 29/10/2021. In addition, the literature search was completed with SPORTDiscus that contains theses and dissertations and thereby included "gray literature" (i.e., literature that is difficult to locate or retrieve [14]). This led to the identification of 4,897 potential studies for the initial inclusion for this study (Fig 1). The review was not registered and all necessary information is provided in the present manuscript.

### Selection criteria

After removing 2,688 duplicates, the titles and abstracts of 2,209 articles were screened using the online program Abstrackr (URL. http://abstrackr.cebm.brown.edu/), an open-source tool for systematic reviews [15]. Among the 2,209 abstracts, 1,473 articles were excluded in the first screening based on the criteria listed below, narrowing the number down to 736 articles for full-text review. Studies were eligible for inclusion if they met the following criteria: (1) they implemented a model to study EIMD of the lower limbs muscles in healthy adults (18–65 years); (2) the outcome measures included isometric and dynamic articular joint torque using an isokinetic dynamometer or load cell force measurement at least two time points (before and 24 or 48 h after exercise) combined with the evaluation of at least one other indirect marker of EIMD (e.g., CK, ROM, DOMS); (3) they did not use recovery techniques or other interventions (e.g., massage, icing, nutritional supplementation); and (4) they reported all necessary data to calculate standardized mean differences (SMD) (i.e., absolute value, standard deviation and sample size). Studies were excluded if they presented duplicated data from a

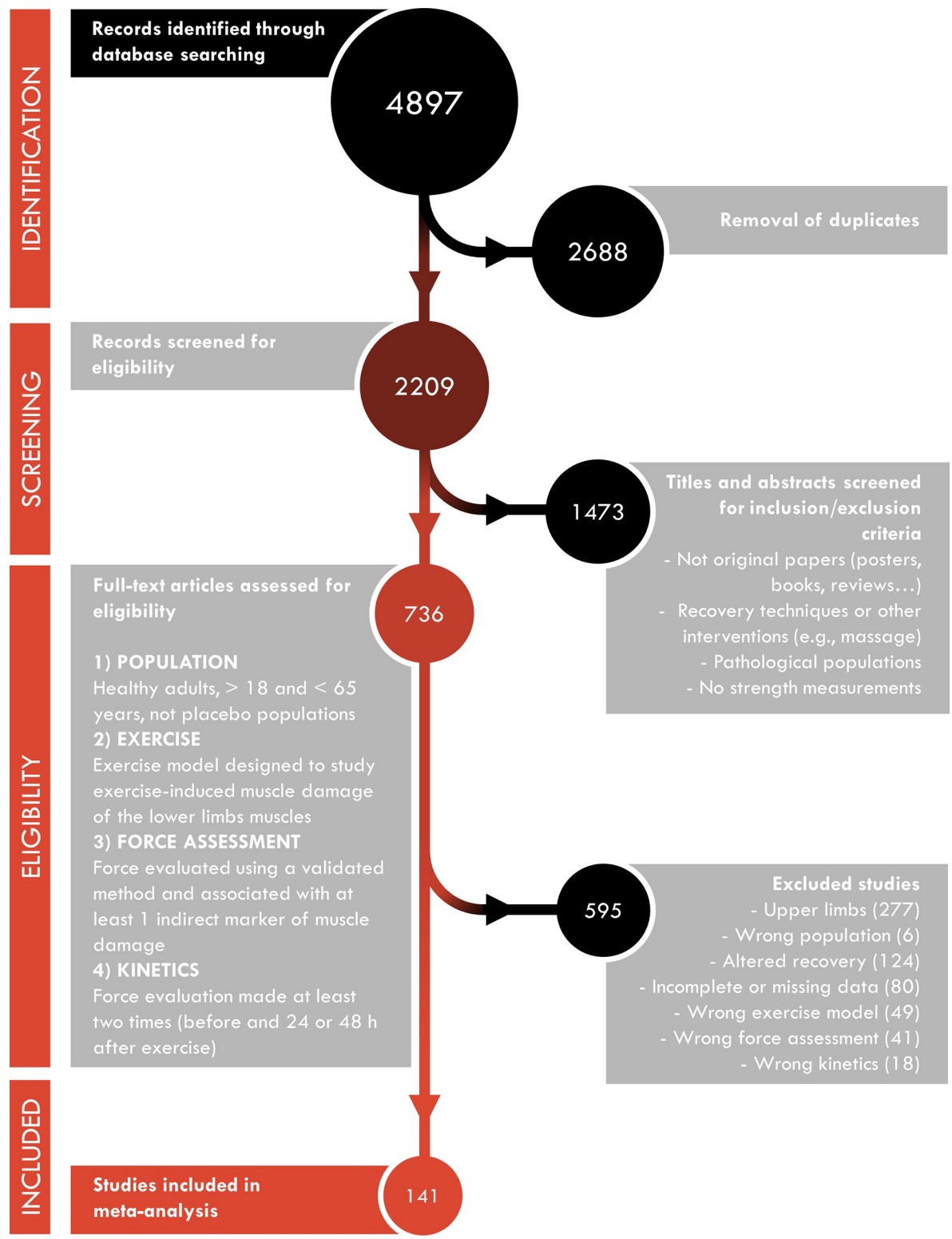

**Fig 1. Schematic flowchart of study selection from initial search to the final study inclusion.**

previous publication. The present study protocol followed the Preferred Reporting Items for Systematic Reviews and Meta-Analyses (PRISMA) statements [16]. The exclusion criteria are displayed in Fig 1.

## Data collection and quality assessment

Included studies (n = 141) were randomly distributed to nine reviewers and data were independently extracted from text or tables using a standardized assessment spreadsheet. The following criteria were collected: sample size, sex, age, training level, exercise type, MVC torque and other indirect markers of EIMD values at all available time points. When incomplete or missing data were found, the corresponding authors of the studies were contacted or the required data was extrapolated from figures using ImageJ software (v1.51j8; [17]). Only EIMD markers present in at least five studies were included in the analysis: jump height, ROM, DOMS assessed actively and passively, pain pressure threshold (PPT), limb circumference, mechanical response to peripheral stimulation (evoked torque response), rate of torque development (RTD), voluntary activation level (VAL), CK, Mb, lactate dehydrogenase (LDH), interleukin-6 (IL-6), and transverse relaxation time ($T_2$) in magnetic resonance imaging (MRI). Due to the large number of studies assessing DOMS using several different methods, they were classified for "active DOMS" (i.e., during a movement), "passive DOMS" (i.e., without movement) and PPT. The quality of the included manuscripts was assessed by the same nine reviewers based on an adapted checklist proposed by Downs & Black [18] (Fig 2; S1 Data).

## Coding for studies

The following moderators were coded: sex (male, female, or both), training status based on the training volume (sedentary, active, trained), exercise modality (eccentric contractions using an isokinetic dynamometer, eccentric/resistance exercise, plyometric exercise, prolonged aerobic exercise, downhill walking/running, neuromuscular electrostimulation, and others) and exercise type (monoarticular, polyarticular). Polyarticular exercises refer to exercises involving movement at multiple joints. Training volume was coded as follow: sedentary (no exercise), active (> 0 h/week and < 3 h/week), and trained (> 3 h/week) based on information available on the articles. The time points of measurements were also coded (<6, 24, 48, 72, >96 h). Early measurements (i.e., between 0 and 6 h) were grouped as "<6 h", due to the lack of consistencies in the literature to assess "post-exercise" EIMD markers. In the same way, the delayed measurements assessed between 96 to 168 h were grouped as "> 96 h" due to the relatively small number of assessments at these time points. Each moderator and time points were coded by two independent reviewers. Any disagreement between reviewers was discussed in a consensus meeting, and unresolved items were addressed by a third reviewer for resolution.

## Statistical analyses

For all analyses, the statistical significance level was set at $p < 0.05$. SMD [19] (also called "effect size" [20]) and standard error (SE) of MVC torque and other indirect markers of EIMD were calculated for each experimental group at several time points of the measurements. The SMD measure can be used to compare the magnitude of changes of similarly computed measures between studies. Indeed, variable changes between studies cannot be simply compared because these studies may differ in their scales/units of measurement or in their standards for precision of measurement [20]. SMDs were weighted by the inverse of the variance to calculate

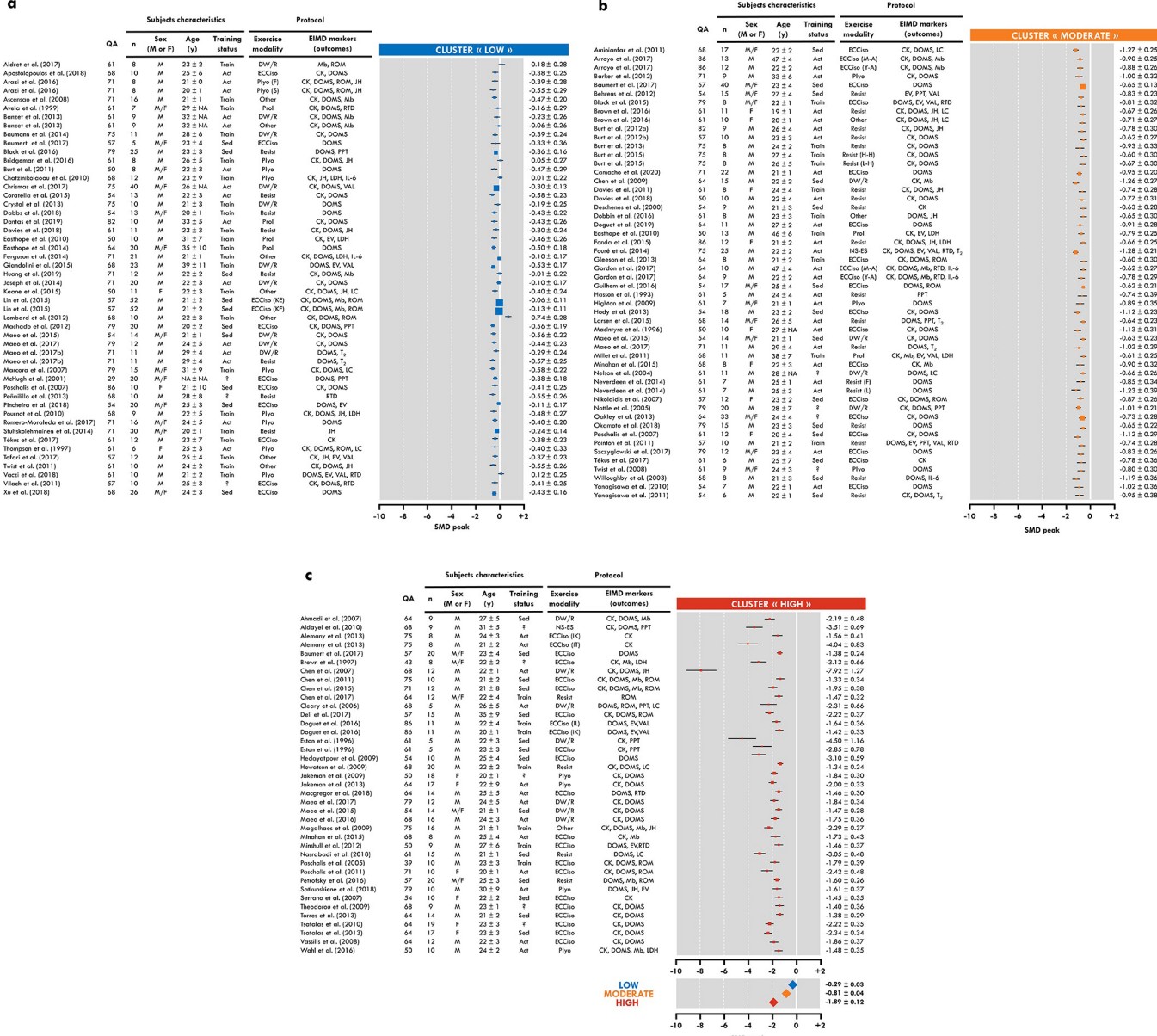

**Fig 2.** Summary of included experimental groups with forest plot for low (a), moderate (b) and high (c) clusters. QA: Quality assessment; EIMD: exercise-induced muscle damage; Sed: Sedentary; Act: Active; Train: trained;?: unknown; ECCiso: eccentric contractions using an isokinetic ergometer; Plyo: Plyometric exercise; Resist: Eccentric/resistance exercises; DW/R: downhill walking/running; Prol: prolonged aerobic exercise; NM-ES: Neuromuscular electrostimulation; CK: creatine kinase; DOMS: delayed-onset muscle damage; Mb: myoglobin; ROM: range of motion; JH: jump height; EV: evoked response; PPT: pain pressure threshold; VAL: voluntary activation level; LDH: lactate dehydrogenase; RTD: rate of torque development; LC: limb circumference; IL-6: iterleukin-6; T₂: transverse relaxation time. "n" represents the number of participants of the experimental group. Data are displayed as standard mean difference (SMD) ± standard error.

the overall effect and SE. SMD was calculated as follows:

$$SMD = \frac{Mean_{TP} - Mean_{PRE}}{Pooled\ SD}$$

where TP was the value at the time-point of interest, PRE was the baseline value and SD was

the standard deviation. Pooled SD was calculated as follows:

$$Pooled\ SD = \sqrt{\frac{(N_{TP} - 1)VARIANCE_{TP} + (N_{PRE} - 1)VARIANCE_{PRE}}{(N_{TP} - 1) + (N_{PRE} - 1)}}$$

where N was the number of participants.

Cohen's criteria were used to interpret the magnitude of the absolute SMD: $<|0.50|$: small; $|0.50|$ to $|0.80|$: moderate; and $>|0.80|$: large [21]. All calculations were carried out with Comprehensive Meta-analysis software (http://www.meta-analysis.com).

A single composite SMD was calculated to estimate the overall effect of changes in MVC torque and other indirect markers at each time point ($<6$, 24, 48, 72, $>96$ h) [22]. A random-effects model was used because the effect of indirect markers might differ according to the type of exercise, training status, sex or other moderators. This analysis was performed in order to describe the time-course of each indirect marker in response to exercise. Changes from baseline (SMD) were considered as significant when 0 was not included in the 95% confidence interval. Similarly, SMD values were significantly different when no overlap was shown between the 95% confidence intervals of the clusters.

A k-medoids clustering (Partitioning around medoids) was done using R (R Core Team; 2020. R: A language and environment for statistical computing. R Foundation for Statistical Computing, Vienna, Austria. URL https://www.R-project.org) and the optimal number of clusters was determined using the NbClust R [23]. The script used to perform the analysis is available at: https://github.com/SirJul/Chalchat-2022—Meta-analysis. This clustering analysis was used to classify 141 experimental groups into 3 categories (clusters) based on the largest reduction in SMD torque at 24–48 h post-exercise (MVC$_{loss\ 24-48h}$) of each group. Thus, experimental groups were classified as low responders (cluster 1; LOW), moderate responders (cluster 2; MOD) and high responders (cluster 3; HIGH) according to the magnitude of MVC$_{loss\ 24-48h}$ (i.e., SMD). Once allocated in the clusters, their respective overall effect (i.e., SMD) in MVC torque and others EIMD markers were calculated at each time point ($<6$, 24, 48, 72, $>96$ h) using a random-effects model, in order to characterize the magnitude and time-course of commonly used indirect markers of EIMD over different levels of muscle function loss. Positive and negative SMD correspond to an increase and a decrease of indirect markers of EIMD after exercise, respectively. The SMD for each cluster was not calculated for PPT, limb circumference, VAL, RTD, LDH, IL-6 and T$_2$, due to the lack of experimental groups in at least one cluster.

A Chi-square test of association was performed using Jamovi (The jamovi project (2020). jamovi (Version 1.2) [Computer Software]. Retrieved from https://www.jamovi.org, Sydney, Australia) to investigate the relationship between the level of MVC$_{loss\ 24-48h}$ (clusters) and the characteristics of participants and exercises (age, sex, training status, exercise type/modality, and muscles involved) represented by the distribution (frequencies) into each cluster (Table 1).

Meta-regressions were performed using Comprehensive Meta-analysis software to investigate the association between the changes in MVC$_{loss\ 24-48h}$ and changes in indirect markers at each time-point ($<6$, 24, 48, 72, $>96$ h) (Table 2). A correlation between MVC$_{loss\ 24-48h}$ and an EIMD marker measured within 6 h post-exercise indicates that this marker could partly predict the MVC$_{loss\ 24-48h}$, whereas a correlation between MVC$_{loss\ 24-48h}$ and an EIMD marker measured between 24 h and 48 h indicates that this marker could be used as a surrogate at these time points. Moreover, correlation between MVC$_{loss\ 24-48h}$ and an EIMD marker measured later than 48 h post-exercise indicates that this marker is able to retrospectively estimate MVC$_{loss\ 24-48h}$.

**Table 1. LOW, MOD and HIGH clusters' characteristics.** Distribution of training status (a), exercise modality (b) and exercise type (c). Data are expressed in percentage of total participants (%). Expected values corresponds to the distribution of each characteristic in the whole sample. ECCiso: eccentric contractions using an isokinetic ergometer; Plyo: Plyometric exercise; Resist: eccentric/resistance exercises; DW/R: downhill walking/running; Prol: prolonged aerobic exercise; NM-ES: Neuromuscular electrostimulation. Any significant association between clusters and characteristics are displayed (*: $p < 0.05$; **: $p < 0.01$).

| (a) | Cluster* | LOW | MOD | HIGH | Expected |
|---|---|---|---|---|---|
| | Sedentary | 20.0% | 26.9% | 35.9% | 26.9% |
| | Active | 30.0% | 46.2% | 33.4% | 36.9% |
| | Trained | 44.0% | 19.2% | 17.9% | 27.7% |
| | Unknown | 6.0% | 7.7% | 12.8% | 8.5% |
| | Total | 100% | 100% | 100% | 100% |
| (b) | Cluster** | LOW | MOD | HIGH | Expected |
| | ECCiso | 22.0% | 38.5% | 56.4% | 37.6% |
| | Plyo | 20.0% | 5.8% | 10.3% | 12.0% |
| | Resist | 16.0% | 38.5% | 10.3% | 22.7% |
| | DW/R | 20.0% | 7.7% | 18.0% | 14.9% |
| | Prol | 8.0% | 3.8% | 0.0% | 4.3% |
| | NM-ES | 0.0% | 1.9% | 2.5% | 1.4% |
| | Other | 14.0% | 3.8% | 2.5% | 7.1% |
| | Total | 100% | 100% | 100% | 100% |
| (c) | Cluster** | LOW | MOD | HIGH | Expected |
| | Polyarticular | 76.0% | 42.3% | 35.9% | 52.5% |
| | Monoarticular | 24.0% | 57.7% | 64.1% | 47.5% |
| | Total | 100% | 100% | 100% | 100% |

# Results

## Identified studies: Characteristics and classification into clusters

A total of 3,089 participants (24.1 ± 3.5 years old) were computed from the included studies (n = 141). The results of the quality assessment of the included studies is provided in Fig 2 and

**Table 2. Coefficient of determination ($r^2$) and p-value of meta-regressions between peak reductions in maximal voluntary contraction (MVC) torque and changes in jump height, range of motion, active DOMS, passive DOMS, pain pressure threshold (PPT), limb circumference, evoked response, rate of torque development (RTD), voluntary activation level (VAL), creatine kinase (CK), myoglobin (Mb), lactate dehydrogenase (LDH), interleukin-6 (IL-6) and transverse relaxation time ($T_2$).** "n" represents the number of studies included in the analysis. Results with a P-value < 0.05 are considered significant and are displayed in bold characters.

| Time | <6h | | | 24h | | | 48h | | | 72h | | | >96h | | |
|---|---|---|---|---|---|---|---|---|---|---|---|---|---|---|---|
| | $r^2$ | p | n | $r^2$ | p | n | $r^2$ | p | n | $r^2$ | p | n | $r^2$ | p | n |
| Jump Height | 0.21 | **<0.001** | 16 | 0.40 | **<0.001** | 19 | 0.00 | 0.21 | 17 | 0.07 | **<0.001** | 13 | - | - | - |
| Range of motion | 0.23 | **0.01** | 12 | 0.00 | 0.36 | 16 | 0.52 | **<0.001** | 16 | 0.02 | 0.23 | 12 | 0.00 | 0.41 | 8 |
| Active DOMS | 0.03 | **0.02** | 45 | 0.00 | 0.55 | 68 | 0.13 | **<0.001** | 76 | 0.16 | **<0.001** | 48 | 0.28 | **<0.001** | 27 |
| Passive DOMS | 0.00 | 0.13 | 26 | 0.00 | 0.71 | 32 | 0.16 | **<0.001** | 40 | 0.22 | **0.02** | 23 | 0.00 | 0.16 | 10 |
| PPT | 0.17 | 0.24 | 6 | 0.13 | **0.02** | 8 | 0.05 | 0.09 | 11 | 0.10 | **0.02** | 6 | 0.00 | 0.35 | 7 |
| Circumference | 1.00 | **<0.001** | 4 | 0.78 | **<0.001** | 7 | 0.82 | **0.01** | 8 | 0.76 | **0.01** | 7 | 0.31 | 0.08 | 4 |
| Evoked response | 0.00 | 0.16 | 15 | 0.88 | **<0.001** | 11 | 0.91 | **<0.001** | 13 | 0.98 | **<0.001** | 8 | 1.00 | **<0.001** | 8 |
| RTD | 0.04 | 0.36 | 10 | 0.51 | **0.02** | 8 | 0.00 | 0.42 | 9 | - | - | - | 0.95 | **<0.001** | 5 |
| VAL | 0.32 | **0.04** | 10 | 0.29 | 0.29 | 7 | 0.00 | 0.42 | 10 | 0.70 | **<0.001** | 5 | 0.00 | 0.83 | 7 |
| CK | 0.02 | 0.08 | 37 | 0.00 | 0.39 | 73 | 0.00 | 0.61 | 80 | 0.21 | **<0.001** | 49 | 0.00 | 0.19 | 33 |
| Mb | 0.47 | **<0.001** | 15 | 0.00 | 0.26 | 15 | 0.09 | 0.30 | 16 | 0.00 | 0.14 | 10 | 0.00 | 0.93 | 7 |
| LDH | 0.00 | 0.52 | 6 | 0.00 | 0.40 | 8 | 0.00 | 0.52 | 8 | 0.00 | 0.77 | 7 | - | - | - |
| IL-6 | 0.00 | 0.46 | 5 | 0.00 | 0.62 | 5 | 0.46 | 0.13 | 5 | - | - | - | - | - | - |
| $T_2$ | - | - | - | 1.00 | **<0.001** | 5 | 1.00 | **<0.001** | 6 | 1.00 | 0.10 | 4 | - | - | - |

S1 Data. Among them, 81% were males and 19% were females. The participants were classified in four training status categories: sedentary (26.9%), active (36.9%), trained (27.7%), and unknown (8.5%). The participants performed different exercises including eccentric contractions using an isokinetic dynamometer (37.6%), eccentric/resistance exercise (22.7%), downhill walking/running (14.9%), plyometric exercise (12.0%), prolonged aerobic exercise (>45 min; 4.3%), neuromuscular electrostimulation (1.4%) and others (e.g., team sports, intermittent exercise, eccentric cycling) (7.1%). These exercises were classified as monoarticular (47.5%) or polyarticular (52.5%). Monoarticular exercises were performed by knee extensors (80.6%), knee flexors (11.9%), plantar flexors (6.0%) and dorsiflexors (1.5%). Among the 3,089 participants, 2,001 participants were assessed for CK activity (64.8%), 1,974 for active DOMS (63.9%), 760 for passive DOMS (24.6%), 501 for Mb concentration (16.2%), 492 for ROM (15.9%), 486 for jump height (15.7%), 352 for the evoked torque response (11.4%), 247 for VAL (8.0%), 241 for PPT (7.8%), 185 for limb circumference (6.0%), 169 for LDH (5.5%), 154 for RTD (5.0%), 149 for IL-6 (4.8%), 52 for $T_2$ in MRI (1.7%).

Measurement time points of each indirect marker in response to exercise are described in Figs 3–5. $MVC_{loss\ 24\text{-}48h}$ (SMD) values of each group are displayed in Fig 2 for the three clusters: low responders (LOW: SMD center = -0.29 ± 0.03, n = 50), moderate responders (MOD: SMD center = -0.81 ± 0.04, n = 52) and high responders (HIGH: SMD center = -1.81 ± 0.12, n = 39). Time-course changes in each indirect marker of the three clusters are described in Figs 4 and 5. SMD for each cluster was not calculated for PPT, limb circumference, VAL, RTD, LDH, IL-6 and $T_2$, due to the lack of experimental groups in at least one cluster.

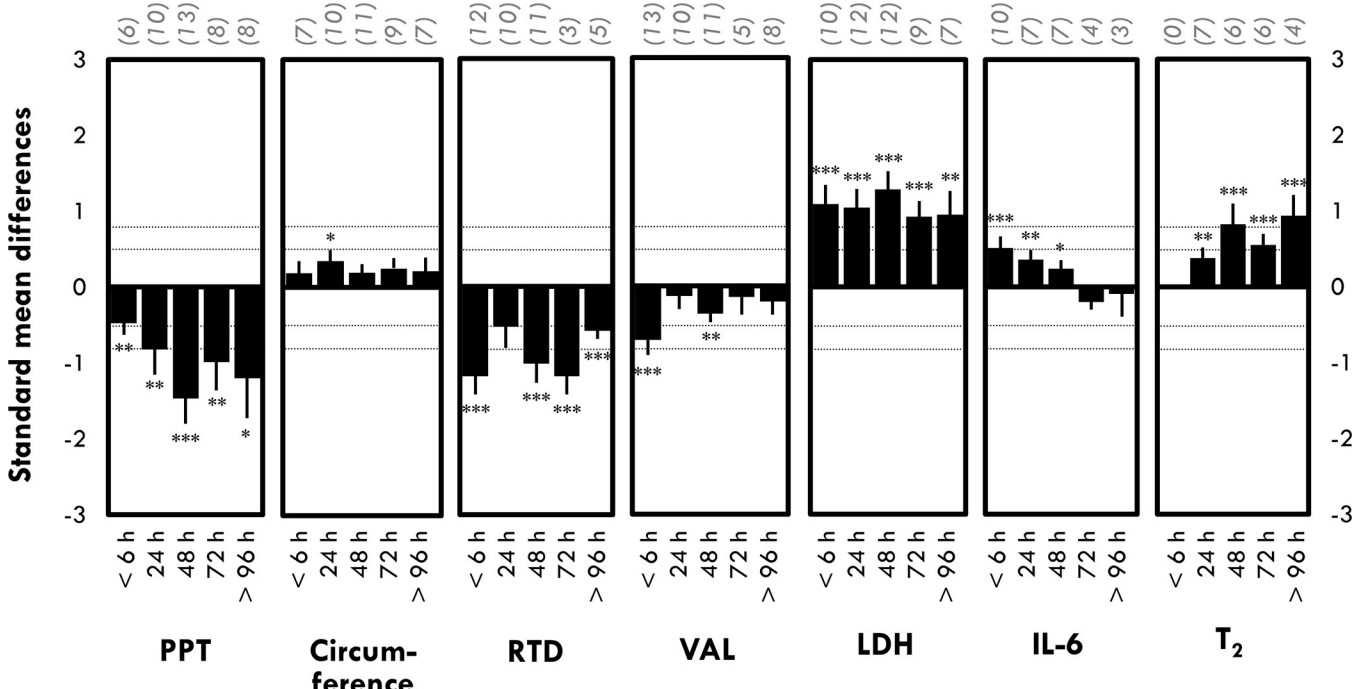

**Fig 3. Time-course changes in pain pressure threshold (PPT), limb circumference, rate of torque development (RTD), voluntary activation level (VAL), lactate dehydrogenase (LDH), interleukin-6 (IL-6) and transverse relaxation time ($T_2$) at < 6 h, 24h, 48h, 72h and > 96 h post-exercise.** The number in parenthesis represents the number of studies. Data are displayed as mean of standard mean difference (SMD) ± standard error. *: Significant difference from baseline at p < 0.05; *; **: Significant difference from baseline at p < 0.01; ***: Significant difference from baseline at p < 0.001.

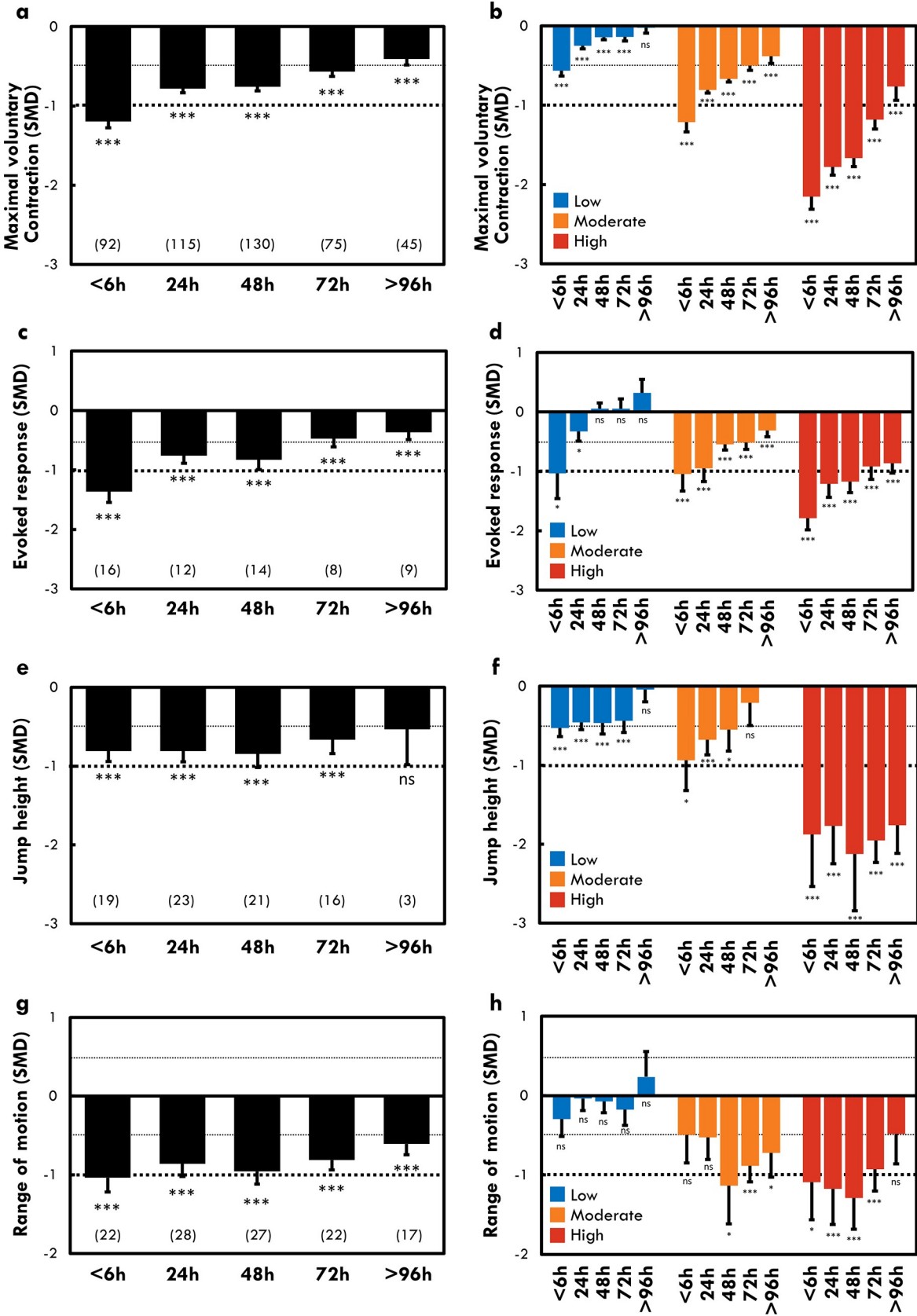

**Fig 4.** Time-course changes in maximal voluntary contraction torque (a-b), evoked response (c-d), jump height (e-f), and range of motion (g-h) after exercise. Standard mean difference (SMD) and standard error were displayed for all included groups (a, c, e, g) and for three clusters representing three levels of maximal voluntary contraction torque loss (b, d, f, h). The number in parenthesis represents the number of studies. Significant differences from baseline (before the damaging exercise) were displayed: *: p < 0.05; **: p < 0.01; ***; p < 0.001; ns: non-significant.

## Changes in EIMD markers: Time-course analysis

**Markers of force-generating capacity.** *MVC torque.* As expected, we found that time-course of MVC torque recovery was different between clusters (Fig 4B). MVC torque recovery was faster in the LOW and MOD clusters when compared with the HIGH cluster (Fig 4B). Indeed, a large effect of exercise on MVC torque (SMD = -1.18 ± 0.12) was still observed at 72 h in the HIGH cluster whereas small to moderate effects were found for the other clusters (LOW: SMD = -0.14 ± 0.05; MOD: SMD = -0.50 ± 0.06).

*Jump height.* Jump height showed greater decrease at 24–48 h post-exercise in the HIGH (SMD = -2.13 ± 0.72) than in the LOW (SMD = -0.47 ± 0.14) cluster after exercise (Fig 4F). Moreover, recovery was faster in LOW and MOD clusters when compared with the HIGH cluster. A large effect of exercise on jump height (SMD = -1.95 ± 0.28) was still observed at 72 h post-exercise in the HIGH cluster, whereas small effects were found for the other clusters (LOW: SMD = -0.44 ± 0.14; MOD: SMD = -0.21 ± 0.29).

*Evoked torque response.* A significant decrease in the amplitude of the evoked torque response was observed during the days following the exercises (Fig 4C). The decrease at 24–48 h post-exercise was larger for HIGH cluster (SMD = -1.21 ± 0.23) compared to the LOW (-0.33 ± 0.16) cluster (Fig 4D). The recovery took longer (> 96 h) in the HIGH cluster than MOD and LOW clusters (48 h) (Fig 4D). Indeed, a large effect of exercise on the amplitude of the evoked torque response was still observed at 48, 72 and 96 h post-exercise, whereas small to moderate and no effects were evident for the MOD and LOW clusters, respectively (Fig 4D).

*Voluntary activation level.* VAL was significantly decreased at <6 h (SMD = -0.71 ± 0.20) and 48 h (SMD = -0.36 ± 0.11) after the exercise (Fig 3). Time-course of this marker was not analyzed for each cluster due to lack of experimental groups in at least one cluster.

*Rate of torque development.* A moderate to large decrease of RTD was observed after the exercise (Fig 3). RTD reduction was the largest within 6 hours following exercise (SMD = -1.18 ± 0.24), transiently recovered at 24 h post-exercise (SMD = 0.53 ± 0.28; p = 0.06) and decreased again at 48 h post-exercise (SMD = -1.01 ± 0.26). Time-course of this marker was not analyzed for each cluster due to lack of experimental groups in at least one cluster.

**Range of motion.** A significant decrease in ROM was observed during the days following the exercises (Fig 4G). However, ROM was significantly reduced only in the MOD and HIGH clusters and the maximum decrease occurred at 48 h post-exercise for these clusters (SMD: -1.14 ± 0.48 and -1.29 ± 0.39, respectively) (Fig 4H). Moreover, ROM was reduced earlier after the end of the exercise in HIGH cluster (<6 h) compare to MOD cluster (48 h).

**Delayed-onset muscle soreness.** Increase in active and passive DOMS and decrease in PPT were observed and peaked 24 to 48 h after exercise (Fig 5A and 5C). DOMS peaked at 48 h post-exercise in the HIGH cluster but peaked at 24 h post-exercise in the LOW and MOD clusters. Moreover, the recovery was faster for the LOW compared to the MOD and HIGH clusters (Fig 5B and 5D). A large effect of exercise on active/passive DOMS was still observed at 72–96 h post-exercise in the HIGH cluster, whereas small to moderate effects were seen for the other clusters at these time points (Fig 5B and 5D).

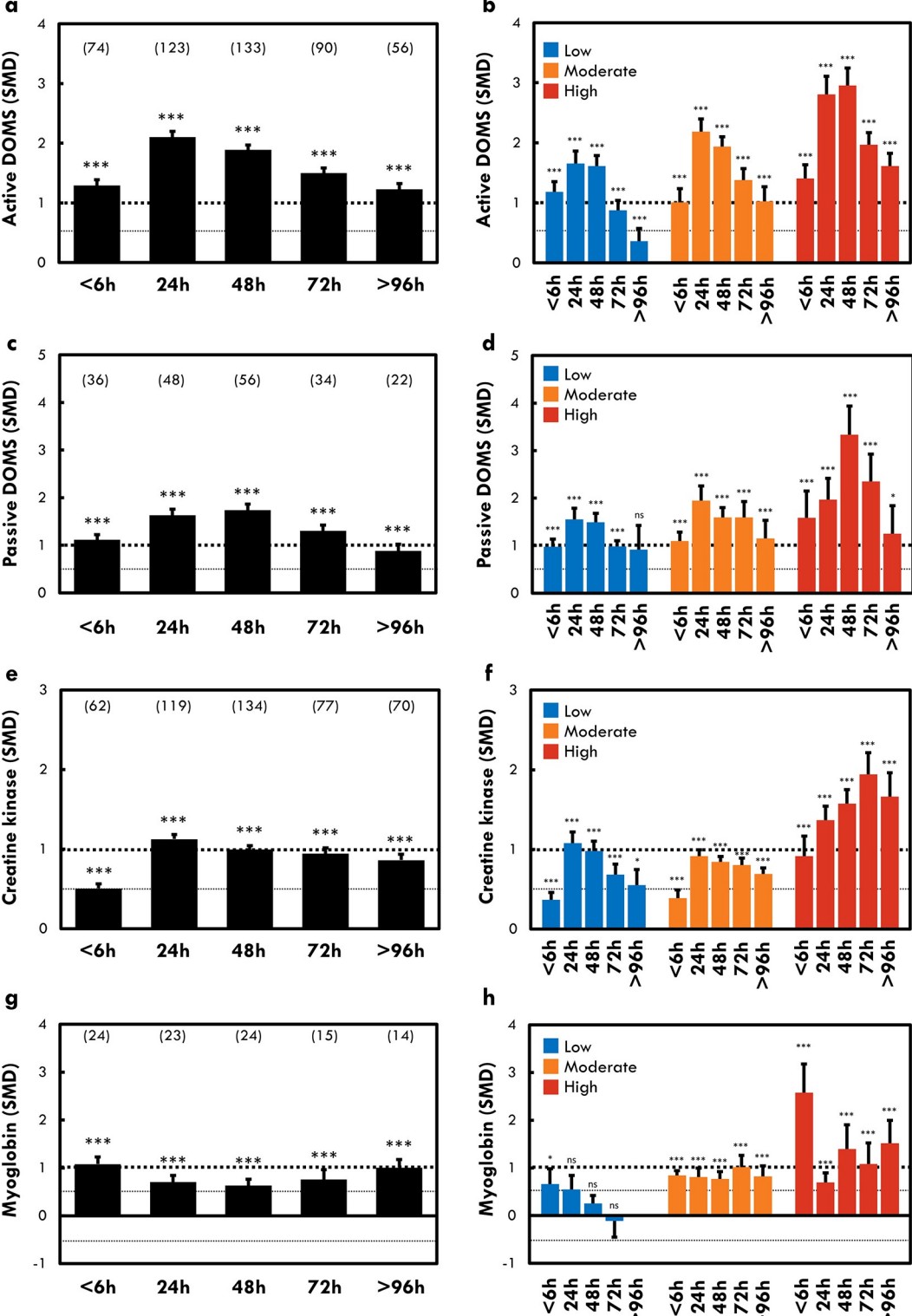

**Fig 5.** Time-course changes in active DOMS (a-b), passive DOMS (c-d), creatine kinase (e-f), and myoglobin (g-h). Standard mean difference (SMD) and standard error were displayed for all included groups (a, c, e, g) and for three clusters representing three levels of maximal voluntary contraction torque loss (b, d, f, h). The number in parenthesis represents the number of studies included in the analysis. Significant differences from baseline (before the damaging exercise) were displayed: *: $p < 0.05$; **: $p < 0.01$; ***; $p < 0.001$; ns: non-significant.

**Swelling.** *Limb circumference.* A small decrease in limb circumference was observed only at 24 h post-exercise (SMD = 0.35 ± 0.15; Fig 3). Time-course of this marker was not analyzed for each cluster due to lack of experimental groups in at least one cluster.

*Transverse relaxation time.* $T_2$ was significantly increased after exercise (Fig 3). Although no study assessed $T_2$ at <6 h post-exercise, larger increases in $T_2$ were observed at 48 to 96 h post-exercise (Fig 3). Time-course of this marker was not analyzed for each cluster due to lack of experimental groups in at least one cluster.

**Blood markers.** *Creatine kinase (CK) activity.* CK activity peaked, on average, 24 h post-exercise and remained elevated for up to 96 h post-exercise (Fig 5E). The time-course changes in CK activity differed as a function of clusters (Fig 5F). CK activity peaked 24 h after exercise for the LOW (SMD = 1.08 ± 0.14) and MOD (SMD = 0.92 ± 0.08) clusters and 72 h post-exercise for the HIGH cluster (SMD = 1.94 ± 0.27) (Fig 5F).

*Myoglobin (Mb) concentration.* Mb concentration peaked, on average, within 6 h and remained elevated for up to 96 h post-exercise (Fig 5G). The time-course of changes in Mb concentration also differed as a function of clusters (Fig 5H). The recovery of Mb concentration was longer in the MOD and HIGH clusters (< 96 h) when compared with the LOW cluster (24 h) (Fig 5H). The maximal Mb concentration was larger for the HIGH cluster (SMD = 2.58 ± 0.60) than the LOW (SMD = 0.66 ± 0.32) cluster.

*Lactate dehydrogenase (LDH).* LDH activity peaked 48 h post-exercise (SMD = 1.28 ± 0.24; Fig 3) remained elevated for up to 96 h post-exercise (SMD = 0.95 ± 0.31; Fig 3). Time-course of this marker was not analyzed for each cluster due to lack of experimental groups in at least one cluster.

*Interleukin-6 (IL-6).* Moderate increase in IL-6 concentration was observed <6 h after exercise (SMD = 0.51 ± 0.15; Fig 3). IL-6 was significantly increased only during the first 2 days following exercise (<6 h, 24 h and 48 h; Fig 3). Time-course of this marker was not analyzed for each cluster due to lack of experimental groups in at least one cluster.

## Factors associated with the MVC$_{loss\ 24-48h}$

The chi-square test of association showed a significant relationship between the level of MVC$_{loss\ 24-48h}$ (clusters) and the training status ($p < 0.05$), the exercise modality ($p < 0.01$) and the exercise type ($p < 0.01$) (Table 1). Trained participants were over-represented in the LOW cluster (44.0%) compared to the expected value (*i.e.*, distribution in the whole sample) (27.7%) (Table 1A). Moreover, eccentric contractions using an isokinetic ergometer and monoarticular exercises were over-represented in the HIGH cluster compared to the expected value (56.4% vs. 37.6% and 64.1 vs. 47.5%, respectively) (Table 1B and 1C).

## Meta-regression: Association between EIMD markers and MVC$_{loss\ 24-48h}$

The results of the meta-regressions between MVC$_{loss\ 24-48h}$ and changes in other markers are shown in Table 2.

**Associations within 6 h post-exercise.** Changes in MVC torque within 6 h post-exercise was significantly correlated ($r^2 = 0.67$; $p < 0.001$) with changes in MVC$_{loss\ 24-48h}$. Jump height, ROM, active DOMS, limb circumference, VAL and Mb concentration changes observed within 6 hours after exercise (SMD value at <6 h) were also correlated with the largest changes in MVC$_{loss\ 24-48h}$ (Table 2).

**Associations between 24 and 48 h post-exercise.** Jump height, PPT, limb circumference, amplitude of the evoked torque response, RTD and $T_2$ changes observed at 24 h post-exercise were correlated with the largest change in MVC$_{loss\ 24-48h}$ (Table 2). Similarly, ROM, active/passive DOMS, limb circumference, amplitude of the evoked torque response and $T_2$ changes

observed at 48 h post-exercise were correlated with the largest change in $MVC_{loss\ 24-48h}$ (Table 2).

   **Associations later than 48 h post-exercise.**   Jump height performance, active/passive DOMS, PPT, limb circumference, evoked torque response, VAL and CK activity changes observed at 72 h post-exercise were correlated with the largest change in $MVC_{loss\ 24-48h}$ (Table 2). Similarly, active DOMS, evoked torque response and RTD measured at >96 h post-exercise were correlated with $MVC_{loss\ 24-48h}$ (Table 2).

## Discussion

This study was the first to quantitatively review the magnitude and time-course of the most common indirect markers of EIMD and cluster them based on levels of $MVC_{loss\ 24-48h}$ following lower limb eccentric-biased exercises. This approach was applied to an extensive dataset from 141 studies employing various exercise modalities and participant characteristics. Firstly, three different time-courses were identified, corresponding to three levels of $MVC_{loss\ 24-48h}$ (LOW, MOD, HIGH) for commonly used EIMD markers (e.g., CK activity, Mb concentration, DOMS, ROM). For instance, magnitude of changes is greater and recovery slower in HIGH cluster compare to other clusters for MVC torque, evoked torque response, jump height, Mb concentration and the peak may also be reached later in HIGH cluster compare to other clusters for DOMS and CK activity. Secondly, we showed that some factors such as training status and exercise type/modality partly explain the level of $MVC_{loss\ 24-48h}$. For instance, eccentric contractions using an isokinetic dynamometer and monoarticular exercises induced larger $MVC_{loss\ 24-48h}$ than other exercises modalities and polyarticular exercises, respectively. Thirdly, some EIMD markers could serve as surrogate (e.g., jump height) to MVC or could be related to (e.g., Mb) the level of $MVC_{loss\ 24-48h}$.

### Changes in EIMD markers: Time-course analysis

   **Markers of force-generating capacity.**   The results of the present study showed that the larger the $MVC_{loss\ 24-48h}$ of the lower limb muscles, the longer the recovery (Fig 4B). This is consistent with Damas et al. [13] that reported time-course differences between three clusters computed from individual MVC torque loss after maximal eccentric exercise of the elbow flexors. The MVC torque recovery time-course is also consistent with a previous review by Paulsen et al. [11] who showed a longer recovery after "severe damage" compared to "mild" or "moderate" EIMD based on the MVC torque loss. Interestingly, we also found that the larger the $MVC_{loss\ 24-48h}$ (i.e., the larger EIMD), the longer the evoked torque response and jump height recovery. Unfortunately, we were unable to compare time-course of VAL and RTD at different levels of $MVC_{loss\ 24-48h}$ due to lack of studies assessing these markers in at least one cluster. However, we found that VAL was decreased during the first hours after the exercise (<6 h) and at 48 h post-exercise. RTD, a measure of ability of the neuromuscular system to rapidly produce torque during MVC, was decreased in the first hours following exercises (<6 h), but also 48 to 96 h post-exercise. As the ability of neural drive to activate muscle (i.e., VAL) was not affected 72 and 96 h after exercise in the present study, it could be speculated that the decreased RTD at these time points was likely due to peripheral factors. Consistently, it has been suggested [24, 25] that early phase RTD reflects neural mechanisms underlying exercise-induced torque loss, while late RTD may describe the same physiological mechanisms as MVC torque loss. The results of evoked torque response and VAL confirmed that MVC torque loss observed the hours/days after an EIMD event are due to peripheral (i.e., muscular) and central (i.e., nervous) alterations of the neuromuscular system. Moreover, we confirmed that recovery

process to reestablish the force-generating capacity after a training session or an exercise intervention must be individually adapted as a function of extent of $MVC_{loss\ 24-48h}$.

**DOMS, ROM, and swelling.** The present review confirmed that both passive and active DOMS peaked between 24 and 48 h after exercise, and showed that the greater the $MVC_{loss\ 24-48h}$, the later the peak of DOMS occurs. Indeed, we found that DOMS peaked at 24 h after the exercise for LOW and MOD clusters, whereas it peaked at 48 h post-exercise for HIGH cluster. These results were not consistent with those reported by Damas et al. [13] who found that DOMS peaked 48 h after eccentric exercise of the elbow flexors, regardless of the level of MVC torque loss. Theses discrepancies could be due to the large type of exercise modalities, muscle groups and study designs included into the analysis in the present review, which differ from the specific model used by Damas et al. [13]. However, it is difficult to fully explain the delayed peak of DOMS after severe EIMD due to a complex etiology of pain [26]. For instance, muscle inflammation due to white cells infiltration (i.e., macrophages) [11] and intramuscular fluid pressure [27] due to muscle swelling could be different as function of the level of $MVC_{loss\ 24-48h}$, inducing different time-courses of DOMS. Our results showed a small increase of limb circumference at 24 h post-exercise, whereas larger $T_2$ MRI imagining elevation was observed 24 to 96 h post-exercise. As Damas et al. [13] reported that eccentric exercise of the elbow flexors induce an increase in limb circumference up to 120 h regardless of the level of $MVC_{loss\ 24-48h}$, it could be speculated that this marker is more sensitive in upper limbs than lower limbs. Our results showed that ROM was not reduced the days after the exercises for the LOW cluster, while ROM decreased greatly at 48 h post-exercise for the MOD and HIGH clusters. Moreover, ROM was decreased within few hours (<6 h, 24 h) following exercise only in the HIGH cluster. In a clinical or sport context, reproducible, cheap, quickly and easy to use evaluations are important in order to provide a rapid and quantitative diagnosis of the impact of the exercise on the whole muscle system functionality. It could be suggested the assessment of limb circumference could be most appropriate and give more information about muscle swelling or EIMD when the damage is induced on the upper limbs muscles [9, 13]. Moreover, it seems that changes in muscle-tendon stiffness and the presence of edema represented by the measure of ROM [28] could be appear earlier (within the hours after exercise) and stay high longer (72 h) after severe EIMD.

**Blood markers.** Contrary to the above-mentioned indirect markers, circulating blood markers provide a whole body systemic information regarding EIMD. Indeed, when muscle fibers are damaged or permeability of sarcolemma is increased, intramuscular components are released into the extracellular space and bloodstream. Therefore, the increase in level of circulating markers has been often used to quantify EIMD [10, 29]. In the present study, an increase in CK and LDH activities and Mb concentrations were observed the days after the exercises, but with different time-courses, as already reported elsewhere [30, 31]. Because CK is cleared by the reticuloendothelial system and Mb is cleared through the liver and kidney [32], differences in time-course could be different among these blood markers. Moreover, Muslimovic et al. [32] reported a new mechanism of clearance through muscle cell endocytosis that could bring a new light to this phenomenon. While we found that peak of CK activity was not necessarily larger in HIGH cluster compared to LOW and MOD clusters the hours after the exercise, we found that CK activity peaked at 24 h post-exercise for the LOW and MOD clusters and at 72 h post-exercise for the HIGH cluster. Even though CK is largely used in the literature and in clinical practice, this marker does not seem to be sensitive to EIMD in the first hours after the damaging exercise. In line with Lippi et al. [31], the time-course of changes in Mb concentration after low EIMD (LOW cluster) showed a rapid increase and return to baseline within hours (<24 h) after the exercises. However, Mb levels could remain elevated for some days after exercise after moderate to severe EIMD (MOD and HIGH). The persistent elevation

in Mb and CK the days after exercise in HIGH cluster could be at least partly explained by a secondary release of the myocyte content of CK and Mb the days after the end of the exercise. Indeed, it has been reported that secondary muscle damage and fiber necrosis could occur due to the loss of $Ca^{2+}$ homeostasis and inflammatory process [11, 33]. The time-course of blood markers could inform on the muscle state during the days after the exercise since the peak of CK activity was delayed (from 24 h to 72 h) and Mb could remain elevated and after severe EIMD. It could be that the recovery of sarcolemma integrity is dependent of the level of EIMD. This has practical implications for the diagnosis of EIMD after exercise by focusing not only on the amplitude of the blood markers but also on their time-course.

**Recommendations.** Some discordance or asynchrony exists among EIMD markers. After severe EIMD (i.e., high level of $MVC_{loss\ 24-48h}$), Mb concentration peaks at 6 h post-exercice, whereas ROM and DOMS peaked at 48h and CK at 72 h post-exercise, suggesting that the presence of high level of EIMD should be highlighted by different markers as function of the timing of measurement.

### Factors affecting $MVC_{loss\ 24-48h}$

Our results showed that the training level was a determinant of the magnitude of $MVC_{loss\ 24-48h}$ (Table 1). Indeed, this is consistent with Ertel et al. [34] who reported that untrained participants experienced greater EIMD symptoms than trained individuals. The present study also showed that eccentric contractions using an isokinetic ergometer resulted in larger $MVC_{loss\ 24-48h}$ when compared to other exercises. Similarly, we found that monoarticular exercises induced larger $MVC_{loss\ 24-48h}$ when compared to polyarticular exercises. This could be explained by the high proportion of monoarticular exercises performed at high intensity over a large range of motion using an isokinetic ergometer. Indeed, it has been shown that exercise intensity [35] and exercise range of motion [36, 37] are dominant factors affecting the magnitude of changes in EIMD markers (e.g. MVC torque loss). Unfortunately, it was impossible to compare the intensity of all the exercise models included in the present study due to the heterogeneity of the exercise types and the lack of information on the intensity of exercise in some studies. Our results regarding monoarticular and polyarticular exercises, were in line with Paulsen et al. [11] who stated that isolated eccentric muscle actions cause larger decreases in muscle function compared with eccentric-biased exercise (e.g., downhill running).

The methodological process of the present study revealed the lack of information regarding the participants and exercise model characteristics in many studies. Several participants' characteristics such as age, sex, height, weight, training volume and type of exercise should be systematically and accurately reported. Similarly, several exercise parameters such as exercise intensity, duration and volume should be more clearly reported. If applicable, it is also important to report the mode and the velocity of muscle contraction, and the joint angle and the range of motion at which the contraction was performed. A better understanding of factors involved in the exercise could be helpful to better anticipate the magnitude of $MVC_{loss\ 24-48h}$ and better adapt recovery strategy and training loads of athletes. For instance, the more the subject is trained, the lower the EIMD and lesser the time to reestablish the muscle function between two mechanical stimuli.

### EIMD markers reflecting $MVC_{loss\ 24-48h}$

In order to evaluate whether EIMD markers could be used as indicators of muscle functional deficit, we compared the responses of these markers with the magnitude of $MVC_{loss\ 24-48h}$ by performing meta-regressions. Our results showed some significant associations between the responses of $MVC_{loss\ 24-48h}$ and other EIMD markers. These results suggest that some EIMD

markers are suitable to partly predict (e.g., Mb) whereas other markers could be used as surrogate (e.g., jump height) to the measurement of $MVC_{loss\ 24-48h}$.

**Markers of force-generating capacity.** A significant correlation was found between $MVC_{loss\ 24-48h}$ and evoked torque response changes at 24, 48, 72 and >96 h post-exercise (Table 2), suggesting that the amplitude of the evoked torque response could be used as a surrogate of the MVC torque without the necessity to perform maximal efforts. A significant correlation was also found between jump height loss at 24 h post-exercise and $MVC_{loss\ 24-48h}$ ($r^2$ = 0.40), suggesting that jump height could be a practical, cheap and reliable alternative to evaluate the magnitude of $MVC_{loss\ 24-48h}$. (i.e., the magnitude of EIMD). However, the early changes (<6 h) of both the evoked torque response and jump height were poorly associated with $MVC_{loss\ 24-48h}$ (Table 2). It is likely that early measurements of muscle function reflect the combined influence of metabolic fatigue and muscle damage [11], therefore explaining the poor correlation with the $MVC_{loss\ 24-48h}$. Indeed, while we observed a better association of MVC torque loss at <6 h with $MVC_{loss\ 24-48h}$ than with early evoked response and jump height (<6 h), early MVC measurement does not seem to fully explain the observed $MVC_{loss\ 24-48h}$ ($r^2$ = 0.67), likely due to metabolic/central fatigue. We found that the VAL decrease within 6 hours after exercises was associated with the $MVC_{loss\ 24-48h}$, suggesting that the (in)-ability to voluntarily activate the motor units in the early phase after the exercise could partly predict the $MVC_{loss\ 24-48h}$ (i.e., indicate the magnitude of EIMD). However, early (<6 h) decrease in VAL is largely affected by muscle fatigue [38], limiting the use of this marker to predict the magnitude of $MVC_{loss\ 24-48h}$. The presence of DOMS, swelling and inflammation at 48 h post-exercise in the present study could also be involved in the reduction of VAL observed 48 h after the exercise due to the sensitivity of group III-IV afferent fibers to nociceptive and mechanical (pressure) stimuli [38]. However, our results showed that the decrease in VAL at 48 h post-exercise was small, limiting its effect on the decrease in muscle function at this time-point. RTD, a measure of the ability of the neuromuscular system to rapidly produce torque during MVC, was decreased in the first hours following exercises (<6 h), but also 48 to 96 h post-exercise. Among these time-point, only a correlation between RTD at 96 h post-exercise and $MVC_{loss\ 24-48h}$ was found. Therefore, the present results were inconsistent with Penailillo et al. [25] who found that RTD at the 100–200 ms slot could be used as an early and sensitive marker of EIMD. These conflicting results could be explained by the different methods (early 0–50 vs late 100–200 ms windows) used to assess RTD. To sum up, jump height loss could be a practical, cheap and reliable alternative to evaluate the muscle function loss in EIMD context. Moreover, evoked torque response could be used as a surrogate of the MVC torque and allow to assess muscle function without the necessity to perform maximal efforts.

**DOMS, ROM, and swelling.** We found that DOMS (active/passive DOMS and PPT) was poorly associated with the $MVC_{loss\ 24-48h}$ ($r^2$ < 0.28), suggesting that DOMS may be a poor indicator of EIMD. On the contrary, a moderate correlation was found between ROM changes at 48 h post-exercise and the magnitude of $MVC_{loss\ 24-48h}$ ($r^2$ = 0.52), suggesting that ROM may be a better indicator of magnitude of EIMD at this time point. Despite the small elevation of limb circumference, associations were found between limb circumference changes at several time points post-exercise and the magnitude of $MVC_{loss\ 24-48h}$ (Table 2). This is consistent with Damas et al. [13] who found a correlation between MVC torque loss and arm circumference after eccentric exercise of the elbow flexors. Moreover, we found an association between $T_2$ increase at 24–48 h post-exercise and $MVC_{loss\ 24-48h}$. Thus, $T_2$ seems to be suitable for evaluating the EIMD only after 24 h post-exercise. Our results should nevertheless be interpreted with caution due to the small number of studies assessing limb circumference and $T_2$ MRI on the lower limbs.

**Blood markers.** Regarding systemic indirect markers of EIMD, only Mb concentration seems to early (<6 h post-exercise) estimate the magnitude of $MVC_{loss\ 24-48h}$ (Table 2). Although CK activity in blood is the most often used marker, it may not be suitable to accurately predict the magnitude of $MVC_{loss\ 24-48h}$. Indeed, this marker partly reflects the magnitude of $MVC_{loss\ 24-48h}$ only 72 h post-exercise. Similarly, the lack of correlation between change in LDH activity and $MVC_{loss\ 24-48h}$ suggests that LDH measurement is not useful for the evaluation of the magnitude of EIMD, likely due to the non-specificity of LDH to myocytes [39]. Despite the small to moderate increase of IL-6 concentration the days after the exercises (Fig 3), this marker was not correlated to the magnitude of $MVC_{loss\ 24-48h}$. This suggests that IL-6 concentration is not useful for the estimation of the magnitude of EIMD. These observations are consistent with those of Paulsen et al. [11] and Reihmane & Dela [40] who reported that discrepant changes in IL-6 concentrations among studies were explained by different exercise protocols and training status of the participants. Moreover, this large variability observed in the marker after eccentric exercise could be due to the various role of IL-6 in response to muscle contractions [41].

**Recommendations.** Among the most common used EIMD markers, Mb concentration or MVC torque loss should be used during the first hours after the exercise (<6 h), even if these markers only partly predict the $MVC_{loss\ 24-48h}$ due to high inter-individual variability and metabolic fatigue. MVC torque loss should be used beyond 24 h after the exercise, however, jump height loss and evoked torque response could be surrogates to this measure at these times points. It should be noted that ROM could be an easy and inexpensive alternative when the measure is performed 48 h after the exercise. Finally, the assessment of CK activity seems to be suitable only at 72 h post exercise to provide information on the muscle function.

## Limitations

A few points may have influenced our results and deserve consideration. Firstly, the use of group values rather than individual values could limit the interpretation of the individual variability. Secondly, the assessment of EIMD markers was not always reported for all measurement time points, which limits the interpretation of the time-course changes in these markers, especially for the less frequently assessed markers. The fact that we reviewed only EIMD markers present in at least five studies and that we cannot characterize the time-course for each cluster for some markers, our analysis was limited to the most common used markers. Thirdly, majority of the studies included mainly males (81%), limiting the interpretation of the present results for females. Indeed, it has been shown that muscle function is less affected in female than male after an eccentrically-biased exercise [42]. The lower susceptibility of female to EIMD could be explained by several factors such as the smaller absolute MVC torque [43] or the role of sex hormones [44], but more studies should be performed in order to clarify this.

## Conclusion

The present review was the first to examine the recovery time-course of the most frequently reported indirect markers of EIMD for different levels of $MVC_{loss\ 24-48h}$. As some discordance or asynchrony exists among the most common used EIMD markers, different markers should be used as function of the timing of measurement. Mb concentration or MVC torque loss should be used during the first hours after the exercise (<6 h), even if these markers only partly predict the $MVC_{loss\ 24-48h}$. Jump height loss and evoked torque response could be used as surrogate of the MVC torque loss. ROM could be an easy and inexpensive alternative when the measure is performed 48 h after the exercise, whereas the assessment of CK activity seems to be suitable only at 72 h post-exercise to provide information on the muscle function.

Moreover, factors such as training status and exercise type/modality (e.g. monoarticular exercise, eccentric contractions using an isokinetic ergometer) could affect the magnitude of $MVC_{loss\ 24-48h}$, and therefore affect the time to reestablish the muscle function between two mechanical stimuli.

## Supporting information

**S1 Data. Quality assessment.**
(XLSM)

**S1 Fig. Schematic flowchart of study selection from initial search to the final study inclusion.**
(TIF)

## Acknowledgments

We sincerely thank all support from all authors. No conflict of interest, financial or otherwise, is declared by the authors.

## Author Contributions

**Conceptualization:** Emeric Chalchat, Anne-Fleur Gaston, Luis Peñailillo, Vincent Martin, Sebastian Garcia-Vicencio, Julien Siracusa.

**Data curation:** Emeric Chalchat, Anne-Fleur Gaston, Keyne Charlot, Luis Peñailillo, Omar Valdés, Pierre-Emmanuel Tardo-Dino, Sebastian Garcia-Vicencio, Julien Siracusa.

**Formal analysis:** Emeric Chalchat, Anne-Fleur Gaston, Sebastian Garcia-Vicencio, Julien Siracusa.

**Methodology:** Emeric Chalchat, Anne-Fleur Gaston, Keyne Charlot, Luis Peñailillo, Vincent Martin, Sebastian Garcia-Vicencio, Julien Siracusa.

**Validation:** Emeric Chalchat, Anne-Fleur Gaston, Keyne Charlot, Luis Peñailillo, Omar Valdés, Pierre-Emmanuel Tardo-Dino, Kazunori Nosaka, Vincent Martin, Sebastian Garcia-Vicencio, Julien Siracusa.

**Writing – original draft:** Emeric Chalchat.

**Writing – review & editing:** Emeric Chalchat, Anne-Fleur Gaston, Luis Peñailillo, Kazunori Nosaka, Vincent Martin, Sebastian Garcia-Vicencio, Julien Siracusa.

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
