## [Decision Letter · Decision Letter 0]

26 Apr 2022

PONE-D-22-07742Appropriateness of indirect markers of muscle damage following lower limbs eccentric-biased exercises: A systematic review with meta-analysisPLOS ONE

Dear Dr. Chalchat,

Thank you for submitting your manuscript to PLOS ONE. After careful consideration, we feel that it has merit but does not fully meet PLOS ONE’s publication criteria as it currently stands. Therefore, we invite you to submit a revised version of the manuscript that addresses the points raised during the review process.

Both reviewers and myself read your paper carefully, and I fully agree with the comments that the reviewers have provided. Overall, the systematic review was well-written. There are, however, some areas can be improved (see the detailed comments from the reviewers). In addition to addressing the detailed comments and make relevant revisions, I'd like to invite the authors to double check the updated 2020 PRISMA guidelines, and to make sure this review follows the new guidelines.

We look forward to receiving your revised manuscript.

Kind regards,

Xin Ye, Ph.D.

Academic Editor

PLOS ONE

Journal Requirements:

Reviewers' comments:

Reviewer's Responses to Questions

**Comments to the Author**

1. Is the manuscript technically sound, and do the data support the conclusions?

Reviewer #1: Partly

Reviewer #2: Yes

2. Has the statistical analysis been performed appropriately and rigorously? 

Reviewer #1: I Don't Know

Reviewer #2: Yes

3. Have the authors made all data underlying the findings in their manuscript fully available?

Reviewer #1: No

Reviewer #2: Yes

4. Is the manuscript presented in an intelligible fashion and written in standard English?

Reviewer #1: Yes

Reviewer #2: Yes

5. Review Comments to the Author

Reviewer #1: General Comments:

I’d like to commend the authors on a well-written manuscript. However, I would like to point out that the PRISMA guidelines used for the manuscript are the 2015 guidelines, which were updated in 2020 (PRISMA (prisma-statement.org). The authors will need to check the reporting to ensure they are in compliance with PRISMA guidelines and update the manuscript accordingly. Additionally, the data availability statement indicates that all data are freely accessible within the manuscript and supporting information files, but there is not a data file provided within the supporting materials. This is also something that is a component of the PRISMA 2020 guidelines (item 27), so the authors should provide files of the raw data used for analysis in supplementary files and/or a public repository. Specific comments below.

Abstract:

I accept the abstract as is.

Introduction:

Page 3, line 54: The sentence beginning with “Despite….” Is confusing. I recommend changing “unaccustomed, strenuous exercise and specially involving lengthening contractions, or with a repetition of stretch-shortening cycles” to something like “lengthening contractions can alter the ultrastructure of the muscle tissue…” to improve clarity.

Methods:

Page 4, line 95: Was this protocol registered prior to data analysis? Per PRISMA guidelines (item 24a-b), please state where the protocol registration can be found (e.g. link to repository) or state that the protocol was not registered.

Page 4, line 95: Were amendments made to the protocol over the course of data aggregation and performance of analysis? Per PRISMA guidelines (item 24c), details of the protocol development process should be provided.

Page 5, line 108: Please explain what is meant by “an accurate device”. Was there a list of devices that were included/excluded? If so, please include what those criteria were.

Page 5, line 113: As I said above, these guidelines are outdated, and the manuscript currently hasn’t addressed all aspects of the updated guidelines. Please revisit this after updating the manuscript according to updated guidelines and citation.

Page 7, line 161: Are the R scripts used to conduct analyses publicly available? Per PRISMA guidelines (item 27), if they are available this should be indicated.

Page 7, lines 174 – 183: Were all analyses (k-medioids clustering, Chi-square test, meta-regressions) performed in R? If so, please clarify this.

Results:

Page 8, line 185: Was the risk of bias assessed for all included studies for outcomes of interest? Please indicate this within the results.

Page 11, line 274: What is meant by certain categories being “over-represented”, e.g. “ trained participants were over-represented in the LOW cluster”. How is this being assessed?

Discussion:

Page 12, line 306: To clarify, what is meant by polyarticular exercises? Are you referring to exercises involving multiarticular muscles (e.g. hamstrings) or exercises including movement at multiple joints? This point is important because if you’re referring to multiarticular muscles, it would seem to contradict previous findings of increased damage in those types of muscles such as the hamstrings. Please clarify.

Page 12, line 307: I’m not sure that the point around using measures as surrogate markers is particularly strong, especially related to evoked torque responses. Any scenario involving evoked torque responses would also by definition have to include the ability to measure torque, at which point partial estimation of torque loss will provide less value than direct assessment. Additionally, it doesn’t seem like the results from the study provide support for estimating MVIC loss based on variables such as myoglobin. While the changes were related, relationships don’t necessarily imply a predictive capacity.

Reviewer #2: Summary:

This meta-analysis describes a beautiful synthesis of the data from 141 studies and analyzed the time-courses of the most common used indirect markers of exercise-induced muscle damage (EIMD) based on the different levels of maximal voluntary contraction torque loss at 24-48h post-exercise (MVCloss24-48h). It also assessed the correlations between MVCloss24-48h and the EIMD markers as well as the association between MVCloss24-48h and factors such as subjects’ training status and exercise modalities. The major findings include the different time-courses of the EIMD markers corresponding to different levels of MVCloss24-48h, different correlations between the EIMD markers and MVCloss24-48h at different time points, and the significant relationship between MVCloss24-48h and the training status and exercise modalities. The authors concluded that different EIMD markers should be used as function of the measurement time. For example, blood myoglobin concentration should be used during the first 6 hours following the exercise whereas ROM and CK should be measured at 48h and 72h post-exercise, respectively. In addition, training status and exercise modalities can affect the magnitude of MVCloss24-48h and thus influence the time courses of the EIMD markers.

Overall, this is a well-conducted meta-analysis which provides valuable guidance in the design of future studies involving EIMD assessment.

Only some clarifications and minor revisions are needed:

1. Redundant data description in Figure 3 and Figure 4&5.

- The time-courses of MVC, Evoked response, Jump height, ROM (Figure 4a,c,e,g), Active DOMS, passive DOMS, Ck, Mb (Figure 5a,c,e,g) are also described in Figure 3. This reviewer found the data displays in Figure 4&5 are better and easier to understand. So, this reviewer would recommend deleting the redundantly displayed data in Figure 3 and combine the leftover parts with those of Figure 4&5.

2. Why only the lower-limb exercise studies were included?

3. Line 132: “ volume (sedentary, active, trained),”

- What are the definitions of being sedentary, active and trained, especially the difference between “active” and “trained” subjects?

4. Terminology inconsistency.

Line 134: “type of exercise (monoarticular, polyarticular)”

- This is inconsistent with Figure 2, in which Exercise Type column seemly means the exercise modality in the texts of line 134. I assume the column title in the figure 2 should be Exercise Modality instead of Exercise Type.

5. Lines 154-155: “Cohen’s criteria were used to interpret the magnitude of the absolute SMD: <|0.50|:small;|0.50| to |0.80|:moderate; and >|0.80|: large”

- In Figure 2, the SMD values in the three clusters (LOW, MODERATE, HIGH) don’t match with the above definition in the above texts. For instance, some SMD values in the LOW cluster were higher than 0.5, same question for the MODERATE cluster.

6. Lines 177-178: “Meta-regressions were performed to investigate the association between the changes in MVCloss24-48h and changes in indirect markers at each time-point”

- The MVCloss24-48h changes of each individual or each study?

7. Lines 208-209: “MVC torque recovery was faster in the LOW and MOD clusters when compared with the HIGH cluster (Figure 4b).”

- In the method or Figure 4b, it was not mentioned what kind the statistical analysis was used to compare between the clusters. In Figure 4b, the significant difference between the time points and the pre-exercise level was shown. Same issue with all the other markers’ data in Figure 4&5.

8. Lines 305-306: “eccentric contractions using an isokinetic dynamometer and monoarticular exercises induced larger MVCloss 24-48h than other exercises modalities and polyarticular exercises”

- Same issue as in item#7. What are the statistical analysis data that the statement was based on?

9. Lines 441-442: “We found that DOMS (active/passive DOMS and PPT) was poorly associated with the MVCloss 24-48h, suggesting that DOMS should not be used to evaluate the magnitude of EIMD.”

- Is this statement correct? In Table 2, active DOMS was significantly associated with MVCloss 24-48h at <6h, 48h, 72h, and >96h post-exercise, which is similar as that of ROM. So, this review is confused about the statement.

10. Typo in Table 2. In the first column in Table 2, RFD should be changed into “RTD”.

11. The “n” in Table 2, Figure 2 &3 needs to be defined. Does the “n” represent the number of subjects or number of studies?

12. What are the numbers in the parenthesis in Figure 4&5? Does each of them represent the number of the studies?

6. PLOS authors have the option to publish the peer review history of their article (what does this mean?). If published, this will include your full peer review and any attached files.

Reviewer #1: No

Reviewer #2: No

---

## [Author Response · Author response to Decision Letter 0]

25 May 2022

R1: PONE-D-22-07742 “Appropriateness of indirect markers of muscle damage following lower limbs eccentric-biased exercises: A systematic review with meta-analysis”

Dear editor/reviewers,

The authors would like to thank the editor and reviewers for their constructive and fair comments. The reviewers’ comments and suggestions are shown below (in Italic). Point-by-point answers to the reviewer’s comments are also provided below. We have copied and pasted the modified sentences from the revised manuscript in this point-by-point responses, and indicated the corresponding page and line numbers of the revised manuscript.

We hope that you will find that your comments and suggestions have been adequately considered and reflected in the revised manuscript. We appreciate your help to review the manuscript again.

Editor

Both reviewers and myself read your paper carefully, and I fully agree with the comments that the reviewers have provided. Overall, the systematic review was well-written. There are, however, some areas can be improved (see the detailed comments from the reviewers). In addition to addressing the detailed comments and make relevant revisions, I'd like to invite the authors to double check the updated 2020 PRISMA guidelines, and to make sure this review follows the new guidelines.

Reviewer #1

General Comments: 

I’d like to commend the authors on a well-written manuscript. However, I would like to point out that the PRISMA guidelines used for the manuscript are the 2015 guidelines, which were updated in 2020 (PRISMA (prisma-statement.org). The authors will need to check the reporting to ensure they are in compliance with PRISMA guidelines and update the manuscript accordingly. Additionally, the data availability statement indicates that all data are freely accessible within the manuscript and supporting information files, but there is not a data file provided within the supporting materials. This is also something that is a component of the PRISMA 2020 guidelines (item 27), so the authors should provide files of the raw data used for analysis in supplementary files and/or a public repository. Specific comments below.

As suggested by the reviewer, we updated the PRISMA checklist according to the 2020 version and we provided raw data as supplemental data (page 5, L.129; page 8, L.195).

Abstract:

I accept the abstract as is. 

We appreciate this comment. No further change has been made.

Introduction:

Page 3, line 54: The sentence beginning with “Despite….” Is confusing. I recommend changing “unaccustomed, strenuous exercise and specially involving lengthening contractions, or with a repetition of stretch-shortening cycles” to something like “lengthening contractions can alter the ultrastructure of the muscle tissue…” to improve clarity.

As suggested by the reviewer, we modified this sentence according to his/her suggestion. Now, it read (page 3, L.54-56): “Despite the benefits of this type of exercise, lengthening contractions can alter the ultrastructure of the muscle tissue and induce exercise-induced muscle damage (EIMD) [9]”

Methods:

Page 4, line 95: Was this protocol registered prior to data analysis? Per PRISMA guidelines (item 24a-b), please state where the protocol registration can be found (e.g. link to repository) or state that the protocol was not registered. 

The protocol for this systematic review was not registered prior to data analysis. As it is not possible to perform this registration retrospectively on PROSPERO, we cannot provide a registration number. Therefore, we have specified in the manuscript that this protocol was not registered prior to data analysis (page 4, L.101): “The review was not registered and all necessary information is provided in the present manuscript.”, as suggested by the PRISMA guidelines. 

Page 4, line 95: Were amendments made to the protocol over the course of data aggregation and performance of analysis? Per PRISMA guidelines (item 24c), details of the protocol development process should be provided. 

We did not register the data analysis, however, no amendments were performed during the analysis process. Thus, the lack of registration should not reduce the quality of the work.

Page 5, line 108: Please explain what is meant by “an accurate device”. Was there a list of devices that were included/excluded? If so, please include what those criteria were. 

We agree with the reviewer that this statement was ambiguous. Therefore, we amended the sentence as follows (page 5, L.109): “the outcome measures included isometric and dynamic articular joint torque using an isokinetic dynamometer or a load cell »

Page 5, line 113: As I said above, these guidelines are outdated, and the manuscript currently hasn’t addressed all aspects of the updated guidelines. Please revisit this after updating the manuscript according to updated guidelines and citation.

We updated the reference 16 according to the 2020 guidelines (page 5, L.115), and we have checked the “PRISMA 2020 checklist”, which has been provided in this submission. 

Page 7, line 161: Are the R scripts used to conduct analyses publicly available? Per PRISMA guidelines (item 27), if they are available this should be indicated. 

The scripts are publicly available. This is now specified on page 7, L.169: “The script used to perform the analysis is available at: https://github.com/SirJul/Chalchat-2022---Meta-analysis.”

Page 7, lines 174 – 183: Were all analyses (k-medioids clustering, Chi-square test, meta-regressions) performed in R? If so, please clarify this. 

Jamovi and Comprehensive Meta-analysis software were used for analyses. This is now clarified in the revised version of the manuscript: “A Chi-square test of association was performed using Jamovi (The jamovi project (2020). jamovi (Version 1.2) [Computer Software]. Retrieved from https://www.jamovi.org, Sydney, Australia)” (page 7, L.180-181) and “Meta-regressions were performed using Comprehensive Meta-analysis software” (page 8, L.185).

Results:

Page 8, line 185: Was the risk of bias assessed for all included studies for outcomes of interest? Please indicate this within the results.

The risk of bias was assessed by “an adapted checklist proposed by Downs & Black (1998)”. (page 5, L.129) (Figure 2; supplemental data 1). As suggested by the reviewer, this is now indicated in the results section (page 8, L.195): “The results of the quality assessment of the included studies is provided in Figure 2 and supplemental data 1.” 

Page 11, line 274: What is meant by certain categories being “over-represented”, e.g. “trained participants were over-represented in the LOW cluster”. How is this being assessed?

One of the aims of this study was to investigate the relationship between the level of MVCloss 24-48h (clusters) and the characteristics of participants (age, sex, training status) and exercise type/modality. The result (page 11, L.278-283) mentioned by the reviewer means that there are more trained participants on the LOW cluster compared to the overall sample. This was assessed by using a Chi-square test of association, comparing the distribution of trained participants in the LOW cluster to the expected values (i.e., the distribution in the whole sample). Minor modifications were added to this paragraph to clarify the results. This section now reads (page 11, L.280-283): “Trained participants were over-represented in the LOW cluster (44.0 %) compared to the expected value (i.e., distribution in the whole sample) (27.7 %) (Table 1a). Moreover, eccentric contractions using an isokinetic ergometer and monoarticular exercises were over-represented in the HIGH cluster compared to the expected value (56.4 % vs. 37.6 % and 64.1 vs. 47.5 %, respectively) (Table 1b and 1c).”

Discussion:

Page 12, line 306: To clarify, what is meant by polyarticular exercises? Are you referring to exercises involving multiarticular muscles (e.g. hamstrings) or exercises including movement at multiple joints? This point is important because if you’re referring to multiarticular muscles, it would seem to contradict previous findings of increased damage in those types of muscles such as the hamstrings. Please clarify.

We agree that this information is important and was lacking in the previous version of the manuscript. In this work, “Polyarticular exercises refer to exercises involving movement at multiple joints”. This has been added (page 5, L.134-135) in the methods section.

Page 12, line 307: I’m not sure that the point around using measures as surrogate markers is particularly strong, especially related to evoked torque responses. Any scenario involving evoked torque responses would also by definition have to include the ability to measure torque, at which point partial estimation of torque loss will provide less value than direct assessment. Additionally, it doesn’t seem like the results from the study provide support for estimating MVIC loss based on variables such as myoglobin. While the changes were related, relationships don’t necessarily imply a predictive capacity.

We agree that assessment of MVC torque is easier to perform than the assessment of evoked torque responses, therefore, MVC torque assessment should be performed rather than evoked torque responses assessment. However, as stated page 16, L.422-423: “the evoked torque response could be used as a surrogate of the MVC torque without the necessity to perform maximal efforts”. We removed “evoked torque response” page 12, L.313-314, as this result was not the most relevant. The sentence now reads: “Thirdly, some EIMD markers could serve as surrogate (e.g., jump height) to MVC “.

We also agree that the relationships do not necessarily imply a predictive capacity, therefore, we tempered our interpretation. The text was amended as follows: “Thirdly, some EIMD markers could serve as surrogate (e.g., jump height) to MVC or could be related to (e.g., Mb) the level of MVCloss 24-48h” (page 12, L.313-315).

 

Reviewer #2

Summary: 

This meta-analysis describes a beautiful synthesis of the data from 141 studies and analyzed the time-courses of the most common used indirect markers of exercise-induced muscle damage (EIMD) based on the different levels of maximal voluntary contraction torque loss at 24-48h post-exercise (MVCloss24-48h). It also assessed the correlations between MVCloss24-48h and the EIMD markers as well as the association between MVCloss24-48h and factors such as subjects’ training status and exercise modalities. The major findings include the different time-courses of the EIMD markers corresponding to different levels of MVCloss24-48h, different correlations between the EIMD markers and MVCloss24-48h at different time points, and the significant relationship between MVCloss24-48h and the training status and exercise modalities. The authors concluded that different EIMD markers should be used as function of the measurement time. For example, blood myoglobin concentration should be used during the first 6 hours following the exercise whereas ROM and CK should be measured at 48h and 72h post-exercise, respectively. In addition, training status and exercise modalities can affect the magnitude of MVCloss24-48h and thus influence the time courses of the EIMD markers.

Overall, this is a well-conducted meta-analysis which provides valuable guidance in the design of future studies involving EIMD assessment. 

We appreciate this reviewer´s kind words. This review and meta-analysis was performed to quantitatively summarize the information available regarding markers of muscle damage in the lower limbs

Only some clarifications and minor revisions are needed: 

1. Redundant data description in Figure 3 and Figure 4&5. 

- The time-courses of MVC, Evoked response, Jump height, ROM (Figure 4a,c,e,g), Active DOMS, passive DOMS, Ck, Mb (Figure 5a,c,e,g) are also described in Figure 3. This reviewer found the data displays in Figure 4&5 are better and easier to understand. So, this reviewer would recommend deleting the redundantly displayed data in Figure 3 and combine the leftover parts with those of Figure 4&5.

As suggested by the reviewer, we removed the redundant data from the figure 3 and we displayed figures similarly to figures 4 and 5. Please see Figure 3, 4 and 5.

2. Why only the lower-limb exercise studies were included?

We chose to focus on lower limb exercises rather than upper-limb exercises due to their involvement and importance on locomotion and daily activities. Moreover, the response of some EIMD markers on upper limbs eccentric exercise was already reported (Damas et al., 2016) in an original study including 286 participants. This meta-analysis was designed to extend this approach to the lower-limb muscles, as no study has previously summarize the responses to eccentric exercise on these muscles.

3. Line 132: “ volume (sedentary, active, trained),” 

- What are the definitions of being sedentary, active and trained, especially the difference between “active” and “trained” subjects? 

We thank the reviewer for this comment since this information was missing and was important to understand how the responses to eccentric exercise was influenced by the training status. We clarified this point (page 6, L.135-137): “Training volume was coded as follow: sedentary (no exercise), active (> 0 h/week and < 3 h/week), and trained (> 3 h/week) based on information available on the articles.”

4. Terminology inconsistency. 

Line 134: “type of exercise (monoarticular, polyarticular)” 

- This is inconsistent with Figure 2, in which Exercise Type column seemly means the exercise modality in the texts of line 134. I assume the column title in the figure 2 should be Exercise Modality instead of Exercise Type. 

This reviewer is correct. We have modified figure 2. Now, it reads “exercise modality”.

5. Lines 154-155: “Cohen’s criteria were used to interpret the magnitude of the absolute SMD: <|0.50|:small;|0.50| to |0.80|:moderate; and >|0.80|: large” 

- In Figure 2, the SMD values in the three clusters (LOW, MODERATE, HIGH) don’t match with the above definition in the above texts. For instance, some SMD values in the LOW cluster were higher than 0.5, same question for the MODERATE cluster.

The aim of the study was to characterize the magnitude and time-course of commonly used indirect markers of EIMD over different levels of muscle function loss at 24-48 h post-exercise. The levels of muscle function loss were not build based on the Cohen’s criteria, but based on a k-medoids clustering (page 7, L.166): “A k-medoids clustering (Partitioning around medoids) was done using R (R Core Team; 2020. R: A language and environment for statistical computing. R Foundation for Statistical Computing, Vienna, Austria. URL https://www.R-project.org) and the optimal number of clusters was determined using the NbClust R [23]. The script used to perform the analysis is available at: https://github.com/SirJul/Chalchat-2022---Meta-analysis. This clustering analysis was used to classify 141 experimental groups into 3 categories (clusters) based on the largest reduction in SMD torque at 24-48 h post-exercise (MVCloss 24-48h) of each group. Thus, experimental groups were classified as low responders (cluster 1; LOW), moderate responders (cluster 2; MOD) and high responders (cluster 3; HIGH) according to the magnitude of MVCloss 24-48h (i.e., SMD). This approach allowed us to divide a set of data into different homogeneous "groups", which is important to sum up the data. As the colors used in figure 3 in the previous version of the manuscript could be ambiguous, we modified. Please see Figure 3.

6. Lines 177-178: “Meta-regressions were performed to investigate the association between the changes in MVCloss24-48h and changes in indirect markers at each time-point” 

- The MVCloss24-48h changes of each individual or each study? 

This sentence was related to the changes reported in each study. As stated in the limitation part (page 18, L.481-482): “Firstly, the use of group values rather than individual values could limit the interpretation of the individual variability”. However, the results showed that this method allowed to characterize the time-course of commonly used indirect markers of EIMD over different levels of muscle function loss at 24-48 h post-exercise.

7. Lines 208-209: “MVC torque recovery was faster in the LOW and MOD clusters when compared with the HIGH cluster (Figure 4b).” 

- In the method or Figure 4b, it was not mentioned what kind the statistical analysis was used to compare between the clusters. In Figure 4b, the significant difference between the time points and the pre-exercise level was shown. Same issue with all the other markers’ data in Figure 4&5.

We thank the reviewer for this comment, as this important information was missing in the previous version of the manuscript. We have now clarified this point in the Methods section (page 7, L.163-165): “Changes from baseline (SMD) were considered as significant when 0 was not included in the 95 % confidence interval. Similarly, SMD values were significantly different when no overlap existed between the 95 % confidence intervals of the clusters.” 

8. Lines 305-306: “eccentric contractions using an isokinetic dynamometer and monoarticular exercises induced larger MVCloss 24-48h than other exercises modalities and polyarticular exercises”

- Same issue as in item#7. What are the statistical analysis data that the statement was based on?

As stated on page 7-8, L.180-184: “A Chi-square test of association was performed using Jamovi (The jamovi project (2020). jamovi (Version 1.2) [Computer Software]. Retrieved from https://www.jamovi.org, Sydney, Australia) to investigate the relationship between the level of MVCloss 24-48h (clusters) and the characteristics of participants and exercises (age, sex, training status, exercise type/modality, and muscles involved) represented by the distribution (frequencies) into each cluster (Table 1).”

This is now also clarified in the Results section (page 11, L.278-283):” The chi-square test of association showed a significant relationship between the level of MVCloss 24-48h (clusters) and the training status (p < 0.05), the exercise modality (p < 0.01) and the exercise type (p < 0.01) (Table 1). Trained participants were over-represented in the LOW cluster (44.0 %) compared to the expected value (i.e., distribution in the whole sample) (27.7 %) (Table 1a). Moreover, eccentric contractions using an isokinetic ergometer and monoarticular exercises were over-represented in the HIGH cluster compared to the expected value (56.4 % vs. 37.6 % and 64.1 vs. 47.5 %, respectively) (Table 1b and 1c).”

9. Lines 441-442: “We found that DOMS (active/passive DOMS and PPT) was poorly associated with the MVCloss 24-48h, suggesting that DOMS should not be used to evaluate the magnitude of EIMD.” 

- Is this statement correct? In Table 2, active DOMS was significantly associated with MVCloss 24-48h at <6h, 48h, 72h, and >96h post-exercise, which is similar as that of ROM. So, this review is confused about the statement. 

Our statements were based on the r² value which is higher for ROM at 48 h (r² = 0.52) than in active/passive DOMS (r² < 0.28). We have clarified this in the manuscript (please see page 17, L.449-453): “We found that DOMS (active/passive DOMS and PPT) was poorly associated with the MVCloss 24-48h (r² < 0.28), suggesting that DOMS may be a poor indicator of EIMD. On the contrary, a moderate correlation was found between ROM changes at 48 h post-exercise and the magnitude of MVCloss 24-48h (r² = 0.52), suggesting that ROM may be a better indicator of magnitude of EIMD at this time point.”

10. Typo in Table 2. In the first column in Table 2, RFD should be changed into “RTD”.

Thank you for picking this typo. We have modified the Table 2 as suggested by the reviewer. Now, this reads “RTD”.

11. The “n” in Table 2, Figure 2 &3 needs to be defined. Does the “n” represent the number of subjects or number of studies?

The “n” represents the number of studies in Table 2 and Figure 3 and the number of participants per experimental group in Figure 2. We have clarified this in the Table/figures legends: ““n” represents the number of studies included in the analysis.” (page 22, L.525) and ““n” represents the number of participants of the experimental group.” (page, 23, L.538).

12. What are the numbers in the parenthesis in Figure 4&5? Does each of them represent the number of the studies?

The number in parenthesis represents the number of studies. We added an explanation on page 23, L.549 and L.554-555: “The number in parenthesis represents the number of studies included in the analysis.”.

---

## [Decision Letter · Decision Letter 1]

27 Jun 2022

Appropriateness of indirect markers of muscle damage following lower limbs eccentric-biased exercises: A systematic review with meta-analysis

PONE-D-22-07742R1

Dear Dr. Chalchat,

We’re pleased to inform you that your manuscript has been judged scientifically suitable for publication and will be formally accepted for publication once it meets all outstanding technical requirements.

Kind regards,

Xin Ye, Ph.D.

Academic Editor

PLOS ONE

Additional Editor Comments (optional):

Reviewers' comments:

Reviewer's Responses to Questions

**Comments to the Author**

1. If the authors have adequately addressed your comments raised in a previous round of review and you feel that this manuscript is now acceptable for publication, you may indicate that here to bypass the “Comments to the Author” section, enter your conflict of interest statement in the “Confidential to Editor” section, and submit your "Accept" recommendation.

Reviewer #1: All comments have been addressed

Reviewer #2: All comments have been addressed

2. Is the manuscript technically sound, and do the data support the conclusions?

Reviewer #1: Yes

Reviewer #2: Yes

3. Has the statistical analysis been performed appropriately and rigorously? 

Reviewer #1: Yes

Reviewer #2: Yes

4. Have the authors made all data underlying the findings in their manuscript fully available?

Reviewer #1: Yes

Reviewer #2: Yes

5. Is the manuscript presented in an intelligible fashion and written in standard English?

Reviewer #1: Yes

Reviewer #2: Yes

6. Review Comments to the Author

Reviewer #1: As before, I would like to commend the authors on a well-written manuscript related to an important topic. I have reviewed their responses to previous reviewer comments and the authors have provided the necessary data and scripts. At this point, I have no further changes for the manuscript.

Reviewer #2: (No Response)

7. PLOS authors have the option to publish the peer review history of their article (what does this mean?). If published, this will include your full peer review and any attached files.

Reviewer #1: No

Reviewer #2: No

---

## [Editor Report · Acceptance letter]

4 Jul 2022

PONE-D-22-07742R1 

Appropriateness of indirect markers of muscle damage following lower limbs eccentric-biased exercises: A systematic review with meta-analysis 

Dear Dr. Chalchat:

I'm pleased to inform you that your manuscript has been deemed suitable for publication in PLOS ONE. Congratulations! Your manuscript is now with our production department. 

Kind regards, 

on behalf of

Dr. Xin Ye 

Academic Editor

PLOS ONE